# Study on Ground Settlement Patterns and Prediction Methods in Super-Large-Diameter Shield Tunnels Constructed in Composite Strata

**Jinlan Li [1], Anyu Liu [1] and Huang Xing [2],***

1   School of Civil Architecture and Environment, Hubei University of Technology, Wuhan 430068, China; lijinlan1999@126.com (J.L.); Liu13296580863@163.com (A.L.)
2   State Key Laboratory of Geomechanics and Geotechnical Engineering, Institute of Rock and Soil Mechanics, Chinese Academy of Sciences, Wuhan 430071, China
*   Correspondence: xhuang@whrsm.ac.cn

**Abstract:** This study focuses on investigating the surface settlement characteristics induced by the construction of a super-large-diameter shield tunnel in composite strata. By utilizing a combination of field monitoring and numerical simulation analysis, the surface settlement patterns encountered during the construction process in horizontally distributed typical soil–rock composite strata were summarized based on the 16.03 m super-large-diameter shield tunnel project in the southerly extension of He'ping Avenue in Wuhan. In addition, the collected data were used to enhance the Peck empirical formula. The results of the study show the following: (1) Significant non-uniform settlement occurs along the tunneling direction when the shield machine passes through soil–rock composite strata. The range of non-uniform settlement is approximately 3.1 times the tunnel diameter (D) in soil sections and 1.9 times the tunnel diameter (D) in rock sections. (2) The impact of composite strata on the maximum settlement is greater than its effect on the settlement trough width, with a larger impact within the soil sections compared to the rock sections. (3) The Peck correction formula, which takes into account the distance between the monitoring cross-section and the composite interface, provides more accurate predictions than the original Peck empirical formula.

**Keywords:** soil–rock composite stratum; revised Peck formula; numerical simulation; on-site monitoring

## 1. Introduction

With the accelerating process of urbanization, the construction of underground transportation facilities such as subways is becoming increasingly popular, and shield tunneling technology has become one of the important means of underground tunnel construction. However, with the continuous increase in tunnel diameter and the diversification of usage scenarios, the problem of surface subsidence caused by shield tunneling construction is becoming more and more prominent. In practical construction, it is necessary to predict and control the surface subsidence caused by tunnel excavation reasonably and strictly.

The research methods for studying surface settlement caused by shield tunneling mainly include empirical formula methods [1–3], numerical simulation methods [4–8], model test methods [9–11], among which the empirical formula methods have advantages such as simple and fast algorithms, selection of different parameters according to geological conditions and different tunnel sections, suitability for preliminary research and quantitative evaluation of surface settlement deformation, and reductions in engineering costs and time. Therefore, they have been widely applied in practical engineering. Based on a large amount of measured data, Professor Peck proposed the cross-sectional Peck formula for surface settlement troughs [12]:

$$S_x = S_{max}\exp\left[-\frac{x^2}{2i^2}\right] \tag{1}$$

Here, $S_{\max}$ is the maximum settlement value of the ground surface above the tunnel after excavation; $i$ is the width of the settlement trough in meters; and $x$ is the horizontal distance between the settlement point of the ground surface and the tunnel axis in meters.

Different scholars have made targeted modifications to the Peck formula through analysis and experiment, in order to make it more applicable to different geological regions and engineering conditions. These modifications include adjusting formula parameters and introducing new correction coefficients. These modifications enable more accurate predictions of surface subsidence caused by tunnel construction, providing important theoretical references for the design and construction of tunnels. Experts and scholars have analyzed the laws of surface subsidence caused by shield construction in different regions of China, and have modified the Peck formula based on the results of their analysis to make it more applicable to regional conditions [13–16]. Ding et al. [17] modified the traditional Peck formula through a regression analysis of measurement data, making it suitable for predicting surface subsidence in soil–rock composite strata with hard layers that have a significant impact on surface subsidence compared with soil–rock composite strata. Hu et al. [18] studied the laws of ground subsidence caused by different construction sequences in a tunnel, and modified the Peck formula to more accurately predict ground subsidence. Fang et al. [19] studied the deformation characteristics of soil caused by the construction of a double-decker shield tunnel and modified the Peck formula to investigate the effect of changes in tunnel depth and ground loss rate on surface subsidence. Zhang et al. [20] modified the width coefficient of the settling trough in the Peck formula based on a large amount of measurement data. Bai et al. [21] studied the laws of surface subsidence in the case of asymmetric geological conditions and proposed a modified Peck formula for asymmetric river terrain conditions. Fang et al. [22] analyzed the surface deformation caused by the construction of a large-diameter slurry shield and obtained the range of key parameters in the Peck formula. Modifying the parameters of the Peck formula for special conditions and geological conditions often cannot meet actual prediction needs and introducing new correction coefficients can often achieve better results. Kang et al. [23] obtained a modified Peck formula for the oblique intersection of shield tunnels with existing tunnels by introducing an angle coefficient and verified the formula using numerical analysis.

Currently, research on the application of the Peck formula in composite strata mainly focuses on the upper and lower stratum composite, that is, when the distribution of physical and mechanical properties of the strata on the excavation surface is uneven, such as the composite form of the upper soil layer and the lower rock strata, as shown in Figure 1. Adjusting the parameters of the Peck formula can meet prediction needs. However, in the case of left–right composite strata, as shown in Figure 2, there will be uneven settlement changes along the excavation direction within a certain range of the strata composite surface, and an unreasonable application of the formula may result in large errors or even mistakes, as the Peck formula cannot predict this kind of uneven subsidence change due to limited parameters. Therefore, this paper uses measured data of surface subsidence caused by the excavation of the Wuhan He'ping Avenue South Extension Tunnel super-large-diameter (16.03 m) shield in composite strata to perform Peck formula regression fitting, and analyzes the measured surface displacement with numerical simulation. The law of surface displacement caused by the construction of the super-large-diameter shield in the composite strata area is obtained, and the Peck formula is modified by introducing the monitoring section to the composite surface distance parameter to rationally predict surface subsidence in composite areas, providing safety assurance and guidance for similar shield tunnel construction in the future.

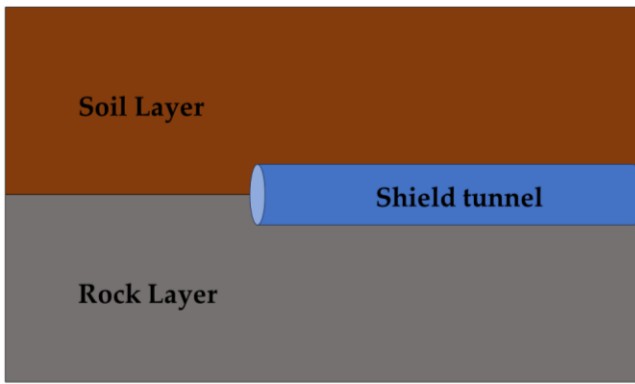

**Figure 1.** Up–down composite stratigraphic diagrams.

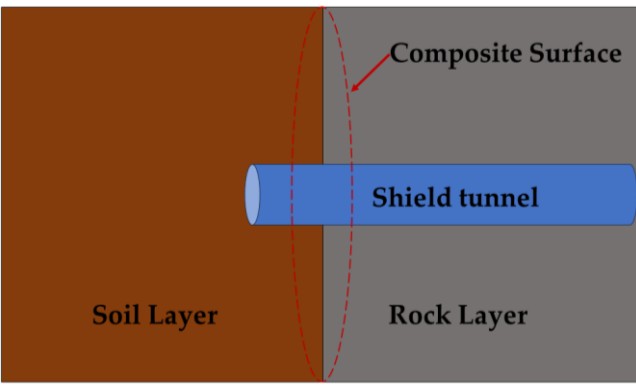

**Figure 2.** Left–right composite stratigraphic diagrams (The red dashed circle is a composite surface.

## 2. Engineering Background

### 2.1. Project Overview

The Wuhan He'ping Avenue South Extension Project has a total length of 3042.5 m, including a tunnel section of 2486 m and a shield section of 1390 m excavated by a 16.03 m diameter slurry balance shield machine. The construction site mainly consists of park landscape areas, municipal roads, and residential areas. Among them, roads with heavy traffic and pedestrian flows such as Wuluo Road, Minzhu Road, Liangdao Street, and Deshengqiao Road, as well as dense underground pipelines, are located along the periphery of the shield section, making the surrounding environment complex and requiring extremely high demands for surface settlement. The terrain units within the traversed zone belong to three levels of terraces: weathered hills, weathered deposits, and the Yangtze River alluvial terraces. The main focus of this study is the traverse area in the Yangtze River alluvial terrace region, with ground elevations along the site ranging from 25.40 to 27.61 m, and the terrain is relatively flat. The traverse path passes through various complex strata, including interbedded clayey silt, limestone, moderately weathered mudstone, and moderately weathered fractured limestone, requiring reasonable predictions of surface settlement to guide safe and efficient construction. The longitudinal section of the strata in the traverse area is shown in Figure 3, and the physical and mechanical properties of the main rock and soil stratum at the site are shown in Table 1.

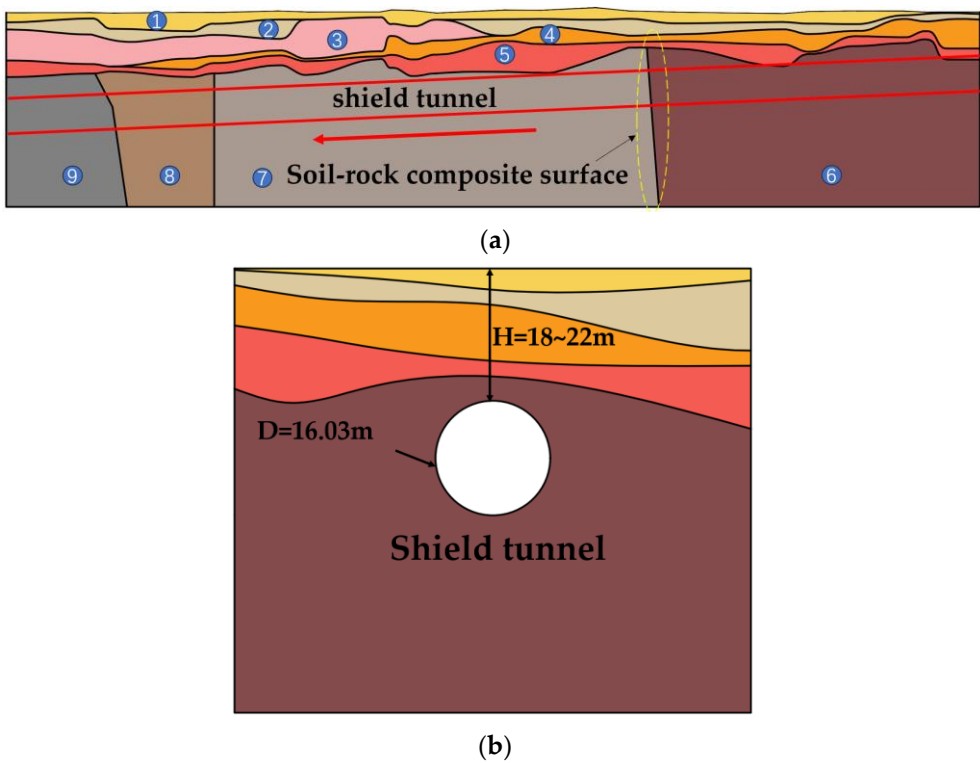

**Figure 3.** Geological profile in shield interval: (**a**) profile view; the number is the stratum number, and the yellow dotted line circle is the soil rock composite surface.and (**b**) cross-section view; the white circle is a shield tunnel.

**Table 1.** Physical and mechanical properties of geotechnical stratum.

| Serial Number | Stratum | $\gamma$/ (kN·m$^{-3}$) | E/MPa | v | c/kPa | $\varphi$/(°) | UCS/MPa |
|---|---|---|---|---|---|---|---|
| 1 | Mixed fill soil | 18.6 | 15.6 | 0.39 | 8 | 10 | |
| 2 | Natural fill soil | 17.5 | 12.0 | 0.37 | 5 | 15.5 | |
| 3 | Silty clay | 19.5 | 25.5 | 0.30 | 25 | 24.5 | |
| 4 | Clay with crushed stones | 19.4 | 36.0 | 0.28 | 62 | 18 | |
| 5 | Red clay | 18.7 | 28.5 | 0.33 | 43 | 17 | |
| 6 | Silty clay with loess | 19.5 | 27.6 | 0.32 | 28 | 10 | |
| 7 | Limestone | 25.7 | 15,900 | 0.25 | 6720 | 42 | 50 |
| 8 | Moderately weathered silty sandstone | 24 | 11,100 | 0.31 | 3000 | 31 | 46.7 |
| 9 | Moderately weathered fractured limestone | 25.8 | 10,690 | 0.32 | 4000 | 39 | 34.4 |

The table includes the following parameters: $\gamma$ for unit weight, E for elastic modulus, v for Poisson's ratio, $\varphi$ for the angle of internal friction, and UCS for unconfined compressive strength.

### 2.2. Installation of Surface Settlement Monitoring Points

The monitoring points were installed by drilling holes at the designated locations using drilling tools and embedding steel bars directly into the relatively solid undisturbed strata. The depth of the embedding is not less than 1.2 m in the undisturbed soil strata and not less than 0.2 m in the disturbed soil strata. The longitudinal settlement monitoring points were set along the axis of the shield driving route. Within a range of 100 m from the starting and ending points of the shield driving section, horizontal settlement slots were set up every 5 m, while in other sections, they were set up every 10 m. The width of the settlement slot monitoring section is 45° to the lower deck of the tunnel roadway. The monitoring points were arranged outward from the centerline of the tunnel at intervals of 4, 4, 4, 4, 6, 6, and 10 m for each section, with a total of nine monitoring points per section. The specific arrangement is shown in Figure 4.

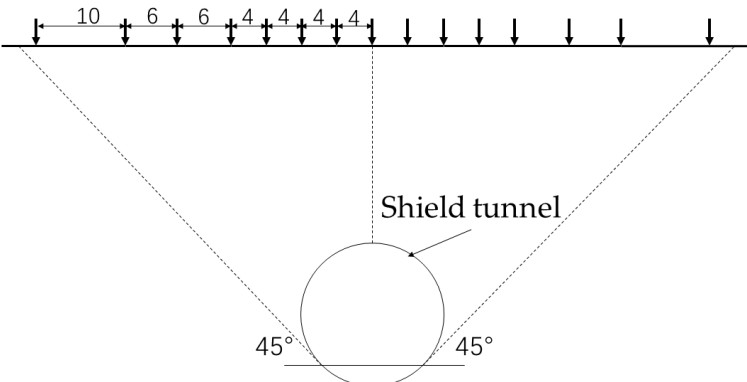

**Figure 4.** Schematic diagram of ground subsidence monitoring point layout (The white circle is a shield tunnel and the arrow is the location of the monitoring point).

## 3. Analysis of Compound Strata and Surface Settlement

Stratum 6 is composed of silty clay interbedded with silt, while stratum 7 is made up of limestone. These two strata exhibit significant differences in their physical and mechanical properties, representing a typical soil–rock compound stratum. Therefore, an analysis was conducted on the section belonging to these strata. Let *L* denote the distance from the monitoring section to the interface between the two strata, as shown in Figure 5. Five sets of monitoring data from each stratum, close to the interface, were selected for analysis. The monitoring data were fitted using the Peck formula to verify its applicability in the compound strata section.

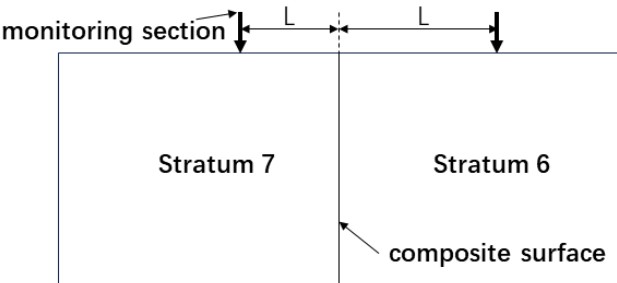

**Figure 5.** Schematic diagram of monitoring section layout (L denote the distance from the monitoring section to the interface between the two strata).

### 3.1. Peck's Empirical Formula

In conjunction with engineering practice, O'Reilly and New derived the relationship expression between the maximum surface settlement $S_{max}$, tunnel excavation radius $R$ in meters, stratum loss rate $V_1$ in percentage, and settlement trough width $i$ in meters. The linear relationship formula between $i$ (m) and tunnel depth $h$ (m) is expressed as follows [24]:

$$S_{max} = \frac{V_1 \pi R^2}{\sqrt{2\pi} i} \tag{2}$$

$$i = k \times h \tag{3}$$

where $k$ is the settlement trough width coefficient.

Based on geological parameters and regional construction experience, the parameter values of the Peck empirical formula are shown in Table 2. Therefore, the Peck empirical formula for stratum 6 and 7 of the stratum can be obtained:

$$\text{Stratum 6}: S_x = 14.02 \times \exp\left[-\frac{x^2}{2 \times 11.44^2}\right]$$

$$\text{Stratum 7}: S_x = 3.8 \times \exp\left[-\frac{x^2}{2 \times 19.36^2}\right]$$

**Table 2.** Parameter values for Peck's empirical formula.

|            | $k$   | $V_\mathrm{l}/\%$ | $i/\mathrm{m}$ | $S_{\max}/\mathrm{m}$ |
|------------|-------|-------------------|----------------|------------------------|
| Stratum 6  | 0.52  | 2.0               | 11.44          | 14.02                  |
| Stratum 7  | 0.88  | 0.53              | 19.36          | 3.8                    |

*3.2. Regression Analysis of the Peck Formula*

We take the logarithm on both sides of the Peck formula, obtaining

$$\ln S(x) = \ln S_{\max} + \frac{1}{i^2} \times \left(-\frac{x^2}{2}\right) \tag{4}$$

$$Y = \ln S(x)$$

$$X = -x^2/2$$

We perform regression analysis using $Y$ and $X$ as the regression variables. Let $\ln S(x)$ be the constant term after regression and $1/i^2$ be the linear coefficient after regression. To simplify notation, we define

$$S_{\mathrm{xx}} = \sum\left(\frac{-x_i^2}{2}\right)^2 - \frac{1}{n}\left(\sum\frac{x_i^2}{2}\right)^2 \tag{5}$$

$$S_{\mathrm{xy}} = \sum\left[\left(\frac{-x_i^2}{2}\right)\ln S(x_i)\right] - \frac{1}{n}\sum\left(\frac{-x_i^2}{2}\right)\sum\ln S(x_i) \tag{6}$$

$$S_{\mathrm{yy}} = \sum\ln^2 S(x_i) - \frac{1}{n}\left[\sum\ln S(x_i)\right]^2 \tag{7}$$

$$b = \frac{S_{\mathrm{xy}}}{S_{\mathrm{xx}}} \tag{8}$$

$$a = \frac{1}{n}\left[\sum\ln S(x_i) - b\sum\left(\frac{-x_i^2}{2}\right)\right] \tag{9}$$

where $a$ is the constant term in the regression equation, $b$ is the linear coefficient in the regression equation, $x_\mathrm{i}$ is the distance from the $i$-th settlement monitoring point to the tunnel centerline, and $n$ is the number of sample points.

We use the $r$-test method in mathematical statistics to determine the goodness of fit, which indicates the closeness of the monitoring values of the variable $\ln S(x)$ to the regression curve. The linear correlation coefficient $r$ is commonly used for this purpose.

$$r = \frac{S_{\mathrm{xy}}}{\sqrt{S_{\mathrm{xx}} \times S_{\mathrm{yy}}}} \tag{10}$$

When $r > 0.8$, $X$ and $Y$ are highly correlated. When $0.5 < r < 0.8$, $X$ and $Y$ are significantly correlated. When $0.3 < r < 0.5$, $X$ and $Y$ have a low correlation. When $r < 0.3$, $X$ and $Y$ are not correlated. Based on this, we obtain the optimal fitting parameters for $S_{\max}$ and $i$ after linear regression:

$$S_{\max} = \exp(a) \tag{11}$$

$$i = \frac{1}{\sqrt{b}} \tag{12}$$

Tables 3 and 4 display the linear regression results for $S_{max}$ and $i$ values, respectively, based on the $r$-values of each monitoring section. Figures 6 and 7 present a comparison between the measured ground settlement data from each monitoring section and the fitting curve of the Peck formula. The following observations can be made:

(1) The linear correlation coefficients ($r$-values) for all monitoring sections are greater than 0.8, indicating a strong correlation with the expected values from Peck's theory. This suggests that the collected monitoring results are reliable, and the fitting curve of the Peck formula accurately reflects the actual ground settlement within the studied project area.

(2) However, there is a significant discrepancy between the measured field values and the predicted results using the Peck empirical formula, indicating that the empirical formula is not capable of providing reasonable predictions for ground settlement in complex geological formations.

**Table 3.** Statistical table of linear regression results for geologic stratum 6.

| L | r | $S_{max}$/m | i/m |
|---|---|---|---|
| 5 m | 0.96 | 9.24 | 16.01 |
| 15 m | 0.99 | 8.5 | 14.8 |
| 25 m | 0.94 | 12.16 | 14.58 |
| 35 m | 0.98 | 12.34 | 13.75 |
| 45 m | 0.99 | 11.78 | 12.78 |

**Table 4.** Statistical table of linear regression results for geologic stratum 7.

| L | r | $S_{max}$/m | i/m |
|---|---|---|---|
| 5 m | 0.97 | 5.58 | 16.78 |
| 15 m | 0.99 | 4.23 | 17.85 |
| 25 m | 0.95 | 3.02 | 18.4 |
| 35 m | 0.95 | 3.67 | 17.9 |
| 45 m | 0.97 | 3.18 | 16.8 |

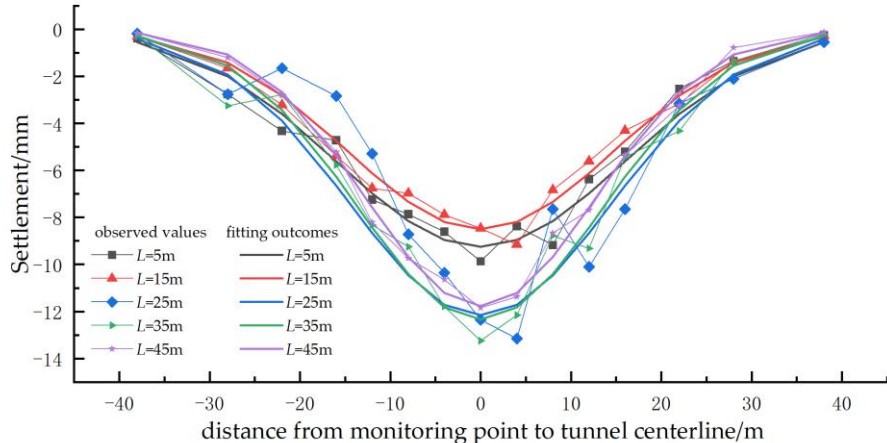

**Figure 6.** Comparison between fitted and monitored ground settlement values for stratum 6.

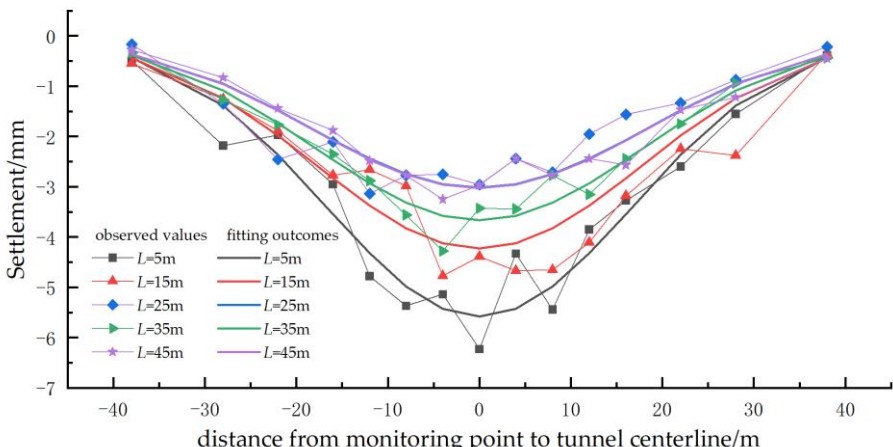

**Figure 7.** Comparison between fitted and monitored ground settlement values for stratum 7.

The widely adopted Abaqus2020 finite element software has found extensive application in engineering research, aiding engineers and scientists in gaining a deeper understanding and facilitating the prediction of the behavior of engineering systems. In order to investigate the uneven settlement range and the variation in $S_{max}$ and $i$ values in the direction of shield tunnel excavations in composite geological formations, the finite element software Abaqus2020 was used to simulate the sixth and seventh geological strata (soil–rock composite formations) in the studied project area.

### 3.3. Numerical Model Establishment

Due to the stress–strain caused by the excavation of underground tunnels, actual effects only exist within a space 3–5 times the diameter of the tunnel excavation around the center of the tunnel. Therefore, the length of the model was set to 150 m along the *X*-axis, the width along the *Y*-axis (driving direction) was set to 200 m, and the height along the *Z*-axis was set to 80 m. Normal displacement constraints were added around the model, full constraints were added at the bottom, and no constraints were added at the top. The shield excavation diameter was set to 16.03 m, the outer diameter of the pipe was set to 15.4 m, the inner diameter was 14.2 m, and the grouting strata thickness was 0.63 m, with a shell thickness of 0.2 m. The three-dimensional numerical model is shown in Figure 8.

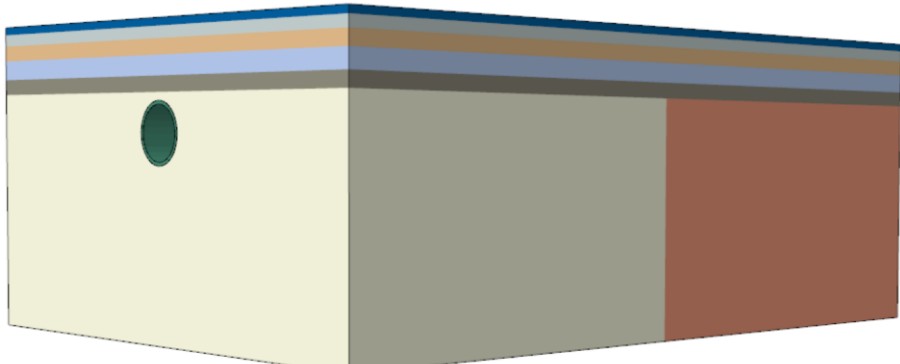

**Figure 8.** Numerical model diagram.

Based on the shield tunnelling parameters collected on site, the synchronous grouting pressure was set to 0.3 MPa, and the face pressure was set to 0.2–0.4 MPa. As shown in Figure 9, the shield tunnelling process was simulated using the model-change function, activating the face pressure and killing excavation soil elements to generate shield shell elements while generating lining and grouting elements at the shield tail and activating

synchronous grouting pressure. This process was repeated for each excavation step, with a length of 2 m per step and a total of 100 excavation steps.

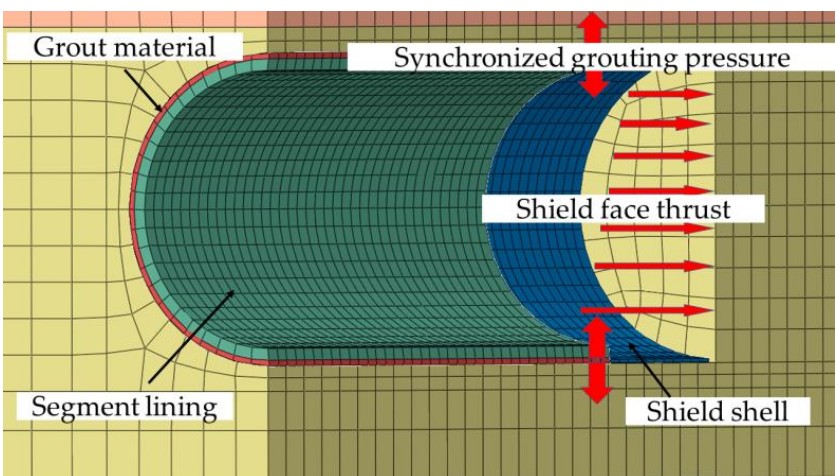

**Figure 9.** Schematic of tunneling parameter settings.

Figure 9 presents a schematic diagram illustrating the parameter settings for shield tunneling in the numerical model, with the mesh division utilizing C3d8r eight-node linear hexahedral elements. The Mohr–Coulomb constitutive model was selected for all soil layers due to its straightforward physical concept, limited parameter set, and its ability to capture the yielding and failure characteristics of geotechnical materials. To account for the specific geological conditions of the project, the physical property parameters for each soil stratum are provided in Table 1. The shield shell, lining segments, and grouting material were modeled using the elastic–plastic constitutive model. As the lining segments were connected by bolts to form a circular structure, the elastic modulus of the segments needed to be multiplied by a rigid reduction factor of 0.8. The hardening process of the grouting material was achieved by controlling the temperature field to vary the elastic modulus and Poisson's ratio. The elastic modulus and Poisson's ratio of the grouting material were 0.9 MPa and 0.4, respectively, when the shield was removed, and increased to 4 MPa and 0.35 after one excavation step and to 800 MPa and 0.25 after two steps. The physical property parameter values for each component are listed in Table 5, respectively.

**Table 5.** Physical and mechanical properties of various components.

| Name | $\gamma$/ (kN·m$^{-3}$) | E/MPa | v |
|---|---|---|---|
| Segment lining | 25 | 34,500 | 0.2 |
| Grout material | 25 | 0.9, 4, 800 | 0.4, 0.35, 0.25 |

The table includes the following parameters: $\gamma$ for unit weight, E for elastic modulus, and v for Poisson's ratio.

Figure 10 illustrates a cloud map of the vertical displacement of the formation after shield tunneling. The results confirm that there is a substantial difference between the surface settlement in stratum 6 in comparison to stratum 7. Near the composite surface of the two formations along the tunneling direction, non-uniform settlements were observed. Considering a numerical model for the analysis, 50 monitoring sections were selected at a 2 m interval ($L$ = 2 m) for each formation interval. The Peck formula regression analysis method (discussed in Section 3.2) was utilized to determine the $S_{\max}$ and $i$ values of each monitoring section based on the simulation results.

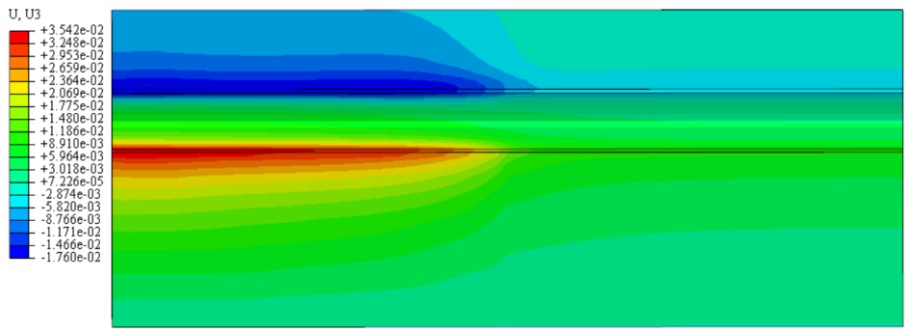

**Figure 10.** Vertical displacement profile of strata after shield tunneling.

Figures 11 and 12 present the numerical simulation results and actual monitoring results of $S_{max}$ and $i$ values for each monitoring section. The fitting degree between the simulation values and the actual monitoring results is high, and the curve trends are essentially the same, indicating the reasonability of the numerical calculation results. It is inferred from the graphs that (1) the surface settlement caused by shield tunneling varies due to the distinct physical and mechanical properties of strata. Therefore, the $S_{max}$ and $i$ values change in a stepwise manner as $L$ varies from soft-rock strata to hard-rock strata within a certain range on both sides of the composite surface, where the maximum surface settlement $S_{max}$ gradually decreases from 15.32 mm to 3.92 mm, and the settlement trough width $i$ gradually increases from 10.25 m to 18.7 m in the direction from soft-rock strata to hard-rock strata. (2) The $S_{max}$ values are significantly impacted by the $L$ distance from the monitoring section to the composite surface in the range of $L = -35 \sim 25$ m ($-2.1D \sim 1.7D$, D is the tunnel excavation diameter), and, similarly, $i$ values are highly influenced by the $L$ distance in the range of $L = -50 \sim 30$ m ($-3.1D \sim 1.9D$). Therefore, we believe that the range of $L = -50 \sim 30$ m ($-3.1D \sim 1.9D$) is the composite strata settlement change range, where the surface settlement amplitudes vary greatly at each section along the direction of shield tunneling, requiring close monitoring during construction. (3) At $L = 0$, the monitoring section is located at the junction of the composite surface of the two strata. The $S_{max}$ value accounts for about 46% of the maximum $S_{max}$ value in interval 6 of the strata and is approximately 1.4 times the minimum $S_{max}$ value in stratum 7 of the strata. Meanwhile, the $i$ value is about 1.5 times the minimum $i$ value in stratum 6 of the strata and accounts for around 88% of the maximum $i$ value in stratum 7 of the strata. In the composite strata settlement change range, the influence of the composite stratum on $S_{max}$ is greater than on $i$, and it has a greater effect on the surface settlement of the soil strata rather than the rock strata.

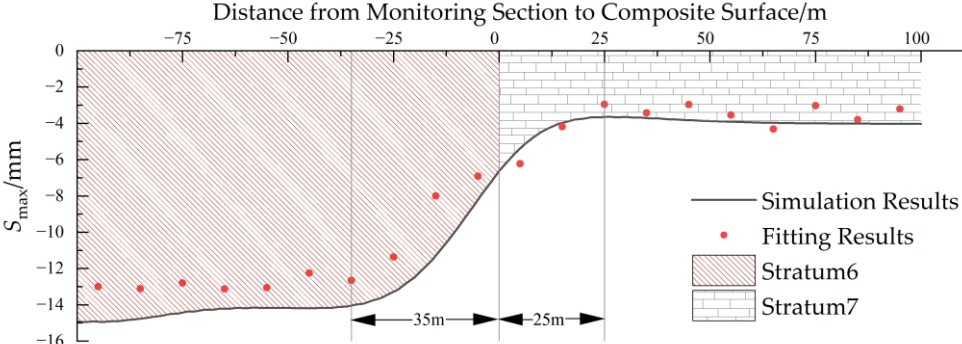

**Figure 11.** Comparison of simulated and measured values of $S_{max}$ for various monitoring cross-sections.

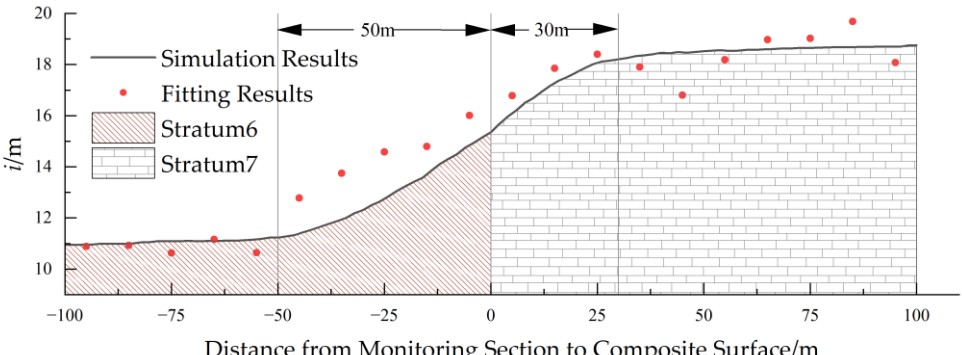

**Figure 12.** Comparison of simulated and measured values of *i* for various monitoring cross-sections.

## 4. Modification of the Peck Formula

Given that Peck's empirical formula only has two characteristic parameters, $S_{max}$ and *i*, it cannot accurately reflect the relationship between surface settlement and the distance *L* between the monitoring section and the composite plane in the settlement change range of the composite strata. Moreover, the values of Smax and *i* are significantly correlated with *L*. Therefore, parameters $\alpha_L$ and $\beta_L$ are introduced to modify the values of $S_{max}$ and *i* in Peck's empirical formula.

$$S_x = S'_{max} \exp\left[-\frac{x^2}{2i'^2}\right] \tag{13}$$

$$S'_{max} = \alpha_L \times S_{max} \tag{14}$$

$$i' = \beta_L \times i \tag{15}$$

Here, $S'_{max}$ and $i'$ represent the modified values of $S_{max}$ and *i*, respectively, and $\alpha_L$ and $\beta_L$ are the modification parameters for $S_{max}$ and *i*. Relationships between various strata's *L* and the modification parameters $\alpha_L$ and $\beta_L$ were obtained through computation and fitting analysis using field measurement data and numerical simulation results.

$$\alpha_L = \begin{cases} \text{Stratum 6}: & 0.65 + 0.01 \times L, & L \leq 35 \\ \text{Stratum 7}: & 1.6 - 0.03 \times L, & L \leq 25 \end{cases}$$

$$\beta_L = \begin{cases} Stratum\ 6: & 1.32 - 0.007 \times L, & L \leq 50 \\ Stratum\ 7: & 0.81 + 0.004 \times L, & L \leq 30 \end{cases}$$

To validate the applicability of the modified Peck formula in the settlement change range of composite strata, a comparison was made between the numerical simulation results and the calculation results obtained using the modified Peck formula. Three examples were considered, with corresponding values of *L* being 5, 15, and 25 meters, as shown in Figures 13 and 14. It can be observed that the predicted maximum settlement values using the Peck empirical formula are significantly larger than the fitted values based on the measured data for the silt–clay interlayer segment, while the settlement values on both sides of the axis are slightly smaller than the fitted values. At *L* = 15 m in the limestone stratum, the predicted results from the Peck empirical formula are slightly smaller than the fitted values, but significant discrepancies still exist at *L* = 5 and 25 m. The predicted values using the modified Peck formula show good agreement with the measured data for both types of strata, achieving satisfactory prediction accuracy for all three example cases. The comparison between the Peck empirical formula and the modified Peck formula indicates that the latter provides more accurate predictions for surface settlement in composite strata.

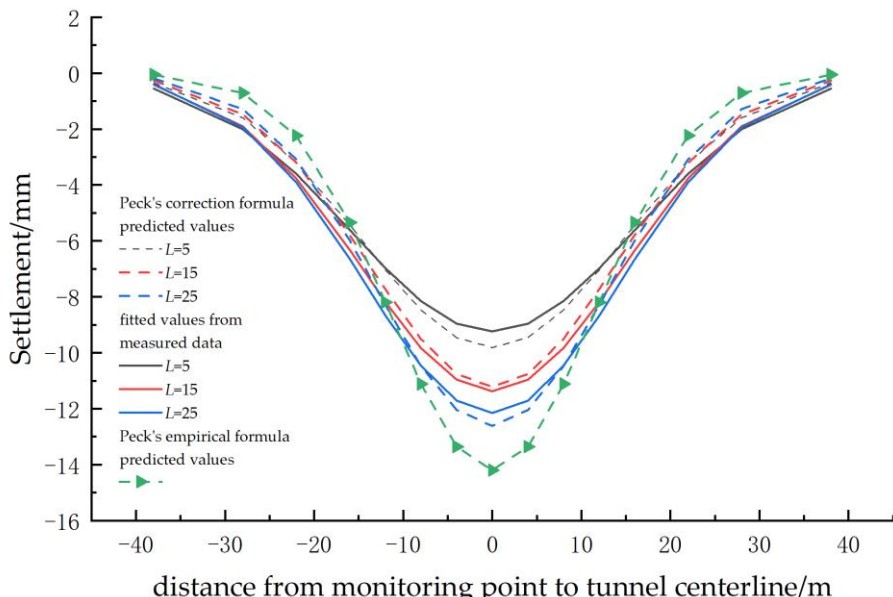

**Figure 13.** Comparison graph of settlement curves among monitoring cross-sections in stratum 6.

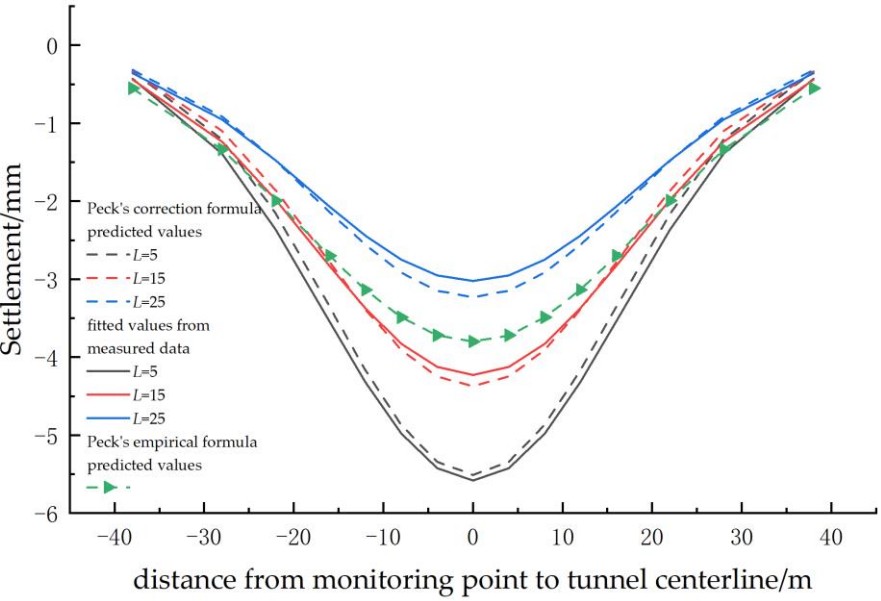

**Figure 14.** Comparison graph of settlement curves among monitoring cross-sections in stratum 7.

## 5. Conclusions and Discussion

Through field tests and numerical analysis, the surface settlement laws and prediction methods of a composite-stratum ultra-large-diameter shield tunnel were studied. The main conclusions are as follows:

(1) When the shield tunnel passes through the soil–rock composite stratum, there will be large and uneven settlement near the composite interface along the direction of the shield excavation. This range is about 3.1D in the soil strata and 1.9D in the rock strata.

(2) The $S_{max}$ of soil strata decreases as the distance to the composite interface decreases, reaching its minimum (46% of the maximum $S_{max}$) at the interface. Conversely, the $S_{max}$ of rock strata increases, reaching its maximum (approximately 1.4 times the minimum $S_{max}$) at the interface. The $i$ value of soil strata as the distance decreases, reaching its maximum (about 1.5 times the maximum $i$) at the interface. Conversely, the $i$ value of rock strata increases, reaching its minimum (88% of the maximum

*i*) at the interface. The composite stratum has a greater influence on $S_{max}$ than *i*, particularly in soil strata.

(3) When predicting the surface settlement of the composite-stratum settlement variation intervals, considering the Peck correction formula for the distance from the monitoring section to the composite interface is more accurate than the Peck empirical formula.

It should be noted that the Peck formula is significantly influenced by engineering conditions. The Peck correction formula provided in this paper is only applicable to the prediction of surface settlement for the Wuhan He'ping Avenue South Extension Tunnel, and it has certain limitations. For the prediction of surface settlement in other regions' shield tunnel projects, it is necessary to readjust the correction parameters based on field measurements. In future studies, a data-driven approach can be explored to improve the adaptability of settlement prediction by leveraging big data.

**Author Contributions:** Conceptualization, J.L.; Writing—original draft, A.L.; Writing—review & editing, H.X. All authors have read and agreed to the published version of the manuscript.

**Funding:** This research received no external funding.

**Institutional Review Board Statement:** The participant's personal identification information used in the study did not include personal information. Ethical review and approval were not required for the study.

**Informed Consent Statement:** Not applicable.

**Data Availability Statement:** Data sharing is not applicable.

**Conflicts of Interest:** The authors declare no conflict of interest.

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
