# Peer review of "Study on Ground Settlement Patterns and Prediction Methods in Super-Large-Diameter Shield Tunnels Constructed in Composite Strata"

_applsci, doi:10.3390/app131910820_

Round 1

Reviewer 1 Report

This paper presents an analytical and numerical study of ground settlement patterns and prediction methods 2 in super-large diameter shield tunnels considering composite geological media. Though paper is generally well-organized and written well, it cannot be published in this present form. The authors need to address the following comments before possible publication.

1. The assertion of uniqueness should have its unique importance properly examined, and any grammatical phrases need to be revised.

2. There is not enough depth in the introduction part regarding the provisions of the present design codal law.

3. Instead of piling up the paper in the literature review, the most recent development should be highlighted. Recommended papers: - Structures, Elsevier, 31, pp. 428-461, DOI: 10.1016/j.istruc.2021.01.102; ;; Buildings, MDPI, 13(5). DOI: 10.3390/buildings13051220.

4. Several parts make certain assumptions. These assumptions need to be supported by arguments. It is important to assess how they will impact the outcomes.

5. What does this paper's major contribution consist of? What drives you to do that? What knowledge deficit in this area has been filled? Which section of the aforementioned instruction has advanced? based on what standards?

6. The text lacked definitions for various variables or symbols that were utilized in some equations.

7. Conclusions drawn by the authors are very long; concise them

8. Accuracy and reliability of the models as well as the employed Peck empirical relations must be clearly highlighted in the study.

9. Last but not least, I must make suggestions resulting from your study. Would you advise revising the government's Regulation or Direction for the design of such underground structures? Is there a need for change?

Recommendation: Major Revision but Acceptable subject to above comments

The paper could be published in the "Applied Sciences" Journal of MDPI if the authors of the manuscript are able to resolve all the specified problems and make the necessary corrections.

Author Response

请参阅附件

Reviewer 2 Report

First of all I appreciate the invitation to review the paper that deals with a Study on Ground Settlement Patterns and Prediction Methods in Super-Large Diameter Shield Tunnels Constructed in Composite Strata.

Here are some comments that are intended to contribute to a better understanding of the content of the work:

1. The authors cite in the abstract of the paper that: (1) Significant non-uniform settlement occurs along the tunneling direction when the shield machine passes through soil-rock composite strata. The range of non-uniform settlement is approximately 3.1 times the tunnel diameter (D) in soil sections and 1.9 times the tunnel diameter (D) in rock sections.

1. Suggestion: the authors could give a brief description of the soil-rock composite stratum in the abstract.

2. Suggestion 2: Still in the abstract, it is suggested to mention more details of shield machine (briefly).

3. Suggestion 3: It is appropriate that the legends in Figures 1 and 2 be reformatted.

4. Question: In view of the tunnel construction method (shield), how does this method interfere in the validation mode of the simulation proposed in this work? Could the authors comment on this aspect of the work?

5. Question: Could the authors comment on the groundwater issue? As there was no citation in the text on this aspect, the water table interferes in what way in the results of the research?

6. Suggestion: for a better understanding of the reader, it would be interesting to include a figure showing the dimensions of the work (Tunnel). Also including the shafts. If there are shafts.

7. Question: In the text of the paper there is no citation of the shafts. If shafts are part of the construction method, how do they interfere with the predictive method?

8. The authors comment that they used the modified peck's empirical formula in the predictive method. Suggestion: it would be interesting to make a brief description of this formula in the initial stage of the paper text to make the work more didactic. 

9. Suggestion: the caption in Figure 3 could have its text improved in terms of the description of the geological cross-section.

10. Suggestion: The dimensions of the work contained in the Project Overview section (2.1) could be transferred to Figure 3. This would be more didactic for understanding the location of the tunnel as well as the geological conditions.

11. The Figure 6. Comparison between Fitted and Monitored Ground Settlement Values for Stratum 6 and Figure 7. Comparison between Fitted and Monitored Ground Settlement Values for Stratum 7 presents absolute values of the comparative measures. The authors do not comment on the errors of these measures.

Question: could the authors comment on the errors and their eventual propagation in the reliability of the measurements?

12. As a suggestion, authors could include an item in the paper devoted to results and discussion. As it stands, where the results and discussions are part of the Engineering Background item, the methodology employed in the study is less valued for the reader.

13. In line 213 of the text of the article, the authors cite that the finite element software Abaqus was used to simulate the 6th and 7th geological layers. Suggestion: the authors could make at this point a brief explanation of what the Abaqus software is.  

14. In line 236 of the text of the article, the authors cite that: According to the actual geological conditions of the project, the Mohr-Coulomb constitutive model was adopted for each layer of soil, with the physical property parameters. Suggestion: the authors could make a brief description of the cited method to make it easier to read.

15. Based on the contents of Figure 9, where the schematic of tunneling parameter settings are shown, each region is separated by a type of geometric mesh. Suggestion: for the reader, it would be easier to understand this situation by a brief explanation of the meaning of each of the borders of these regions.

16. Suggestion: Figures 11 and 12 could be formatted by increasing the font size. This would make it easier to see the predicted and field outcome points.

17. In line 290 of the text of the paper the authors comment that: Given that Peck's empirical formula only has two characteristic parameters, Smax and i, it cannot accurately reflect the relationship between surface settlement and the distance L between the monitoring section and the composite plane in the settlement change range of the composite strata. Question: because it is an empirical relationship, I would like the authors to comment on the limit of adaptation of Pecks' formula?

18. In line 312 of the text of the paper the authors comment that: To validate the applicability of the Peck's modified formula in the settlement change range of composite strata, a comparison was made between the numerical simulation results and the calculation results obtained using the modified Peck formula. Three examples were considered, with corresponding values of L being 5, 15, and 25meters, as shown in Figure 13 and 14. It can be observed that the predicted maximum settlement values using the Peck empirical formula are significantly larger than the fitted values based on the measured data for the silt-clay interlayer segment, while the settlement values on both sides of the axis are slightly smaller than the fitted values. At L=15 meters in the limestone stratum, the predicted results from the Peck empirical formula are slightly smaller than the fitted values, but significant discrepancies still exist at L=5 and 25 meters. The predicted values using the modified Peck formula show good agreement with the measured data for both types of strata, achieving satisfactory prediction accuracy for all three example cases. Question: The authors could comment on the meaning of the statement: the modified Peck formula show good agreement with the measured data for both types of strata?

19. Suggestion: In the conclusions, the authors could estimate the extent to which the predictive method developed in the work can be applied to the varied dimensions of the tunnels.

The quality of writing in the text of the paper can be considered adequate considering the technical aspects of the subject related to the engineering of tunnel construction.

Eventually, authors could turn to an English-language reviewer who has knowledge of terms common to construction engineering.

Round 2

Reviewer 1 Report

I went carefully through the authors' responses to my initial comments as well as to the comments by other reviewers. I also read in detail the revised manuscript proposed by the authors. I think that the authors did provide reasonable answers to my comments. Thank you very much.

I am delighted to review this paper. I hope my comments and suggestions can help improve this work, and I recommend its publication in its current form.